# A Dual-Branch Autoencoder Network for Underwater Low-Light Polarized Image Enhancement

Chang Xue, Qingyu Liu, Yifan Huang, En Cheng and Fei Yuan *

Key Lab of Underwater Acoustic Communication and Marine Information Technology (Xiamen University) Ministry of Education, Xiamen University, Xiamen 361102, China; 23320211154191@stu.xmu.edu.cn (C.X.); 23320231154391@stu.xmu.edu.cn (Q.L.); yfhuang@stu.xmu.edu.cn (Y.H.); chengen@xmu.edu.cn (E.C.)
* Correspondence: yuanfei@xmu.edu.cn

**Abstract:** Underwater detection faces uncomfortable illumination conditions, and traditional optical images sensitive to intensity often cannot work well in these conditions. Polarization imaging is a good solution for underwater detection under adverse lighting conditions. However, the process of obtaining polarization information causes it to be more sensitive to noise; serious noise reduces the quality of polarized images and subsequent performance in advanced visual tasks. Unfortunately, the flourishing low-light image enhancement methods applied to intensity images have not demonstrated satisfactory performance when transferred to polarized images. In this paper, we propose a low-light image enhancement paradigm based on the antagonistic properties of polarization parameters. Furthermore, we develop a dual-branch network that relies on a gradient residual dense feature extraction module (GRD) designed for polarized image characteristics and polarization loss, effectively avoiding noise introduced during the direct amplification of brightness, and capable of restoring target contour details. To facilitate a data-driven learning method, we propose a simulation method for underwater low-light polarized images. Extensive experimental results on real-world datasets demonstrate the effectiveness of our proposed approach and its superiority against other state-of-the-art methods.

**Keywords:** polarization imaging; low-light image enhancement; underwater image enhancement

## 1. Introduction

Underwater optical images serve as a primary means of acquiring information in various underwater applications such as underwater archaeology, underwater infrastructure inspection, and marine biology research. In practical underwater detection, challenging lighting conditions are an unavoidable obstacle. Inadequate illumination or signal attenuation during propagation can lead to an insufficient number of photons received by detectors, which results in intensity image degeneration such as brightness reduction, contrast decrease, and the target information being drowned in noise. Especially regarding exploring in the deep sea, additional artificial lighting may disturb marine life, even though additional lighting is possible, light spots pose a significant challenge to imaging quality. Inspired by mantis shrimp vision, the introduction of polarization information has made a breakthrough in obtaining information in strongly scattering media [1]. Current low-light image enhancement methods mostly rely on intensity and spectral information. In comparison to intensity images, polarized images can still record rich object surface details such as shape, surface roughness, and curvature even in low light or low contrast conditions. According to Figure 1b, it is a fact that the degree of linear polarized image (DoLP) can display target information in low-light conditions. However, it is important to note that the overall image noise is significant, and the quality of the imaging is undeniably low, which is attributed to the heightened sensitivity of polarization parameters to noise. Extracting polarization information from the image involves calculating the Stokes parameters using

intensity measurements and nonlinear calculations. This process introduces complex noise, resulting in polarized images with substantial noise in low-light conditions. Therefore, obtaining high-quality polarized images is crucial for enhancing underwater detection capabilities and obtaining valuable underwater information.

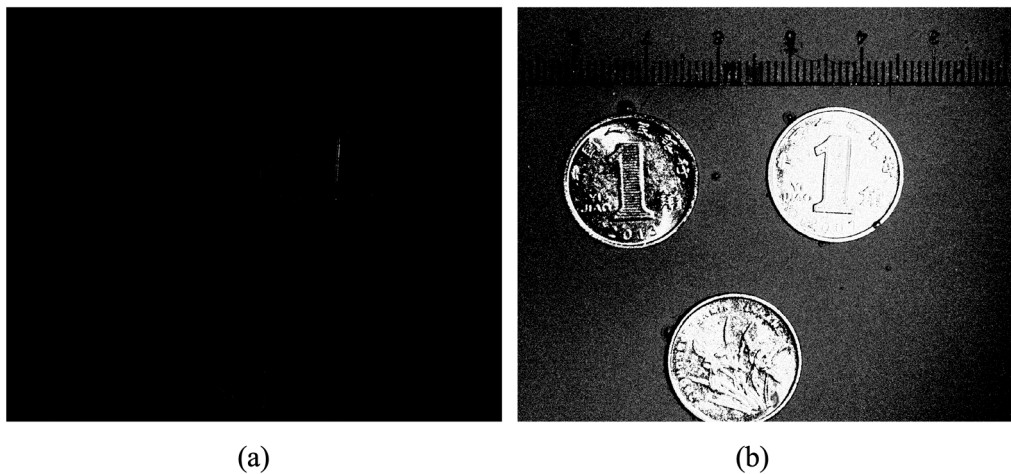

(a) (b)

**Figure 1.** Comparison of intensity and polarized images in low-light conditions: (**a**) intensity image; (**b**) DoLP image.

Existing efforts in low-light image enhancement mainly focus on enhancing individual intensity images and can be categorized into traditional methods and learning-based methods. Traditional methods may suffer from inadequate or excessive enhancement, and further amplification of noise in practical applications [2,3]. Learning-based methods [4–9] utilize deep neural networks to extract image features from a large amount of data to enhance the quality of optical images in the intensity domain. However, when these methods are transferred to the polarized images, their performance significantly deteriorates, which is attributed to the different noise distributions of different domain images. These methods fail to leverage the physical constraints between multiple polarized images, leading to the loss of some physical prior information. Moreover, denoising methods in deep learning may disrupt the original noise relationships, causing more severe degradation when calculating polarization parameters. Some specialized works for polarized images exist as well. Hu et al. [10] propose a method to simultaneously enhance intensity and polarized images by leveraging relationships between multiple polarized images, but it still focuses on processing in the intensity domain. In [11], a low-light enhancement method in the Stokes domain is proposed, and it is found that the degradation of Stokes parameters is much less severe than that of polarized images under low-light conditions. However, directly amplifying brightness will further amplify noise. In addition, there is also no publicly available polarization image dataset, which greatly limits data-driven methods.

To address these limitations, we propose a solution for enhancing underwater low-light polarized images. Inspired by the work [11], we tackle the problem of polarized image enhancement in the Stokes domain. Through extensive observation and analysis of a substantial number of underwater polarized images obtained in real-world experiments, we design a dual-branch network combining the physical priors to address the challenges of reduced brightness and complex noise. To drive the network training, it is necessary to obtain a substantial dataset of paired underwater polarized images. Even with the assistance of polarization cameras, collecting a large and abundant dataset is time-consuming and labor-intensive. There is currently no publicly available underwater polarization image dataset. Therefore, we consider using a simulated dataset to drive network training. Integrating both underwater degradation and low-light degradation into polarization information in a coherent manner to create a comprehensive paired dataset poses a significant challenge. Existing methods for low-light simulation or underwater style transfer are mainly designed for independent images. We aim to develop a more

realistic data synthesis pipeline without compromising the physical constraints relationship between multiple polarized images. Ultimately, we generate a paired underwater low-light polarized image dataset encompassing various water degradation scenarios and diverse lighting conditions. To validate the generalization performance, we also set up an underwater polarimetric imaging system, creating a real-world dataset. We then test the performance of our algorithm on the real-world dataset, demonstrating the robustness and effectiveness of our approach. The main contributions can be summarized as follows:

- We propose a Stokes-domain underwater low-light polarized image enhancement paradigm inspired by a polarization antagonistic relationship, which can effectively avoid the damage caused by directly amplifying brightness and can restore the details by using the physical prior. To our best knowledge, this paradigm designed according to mutual constraints of Stokes parameters is proposed for the first time and demonstrates superior performance compared to other methods.
- Based on the proposed paradigm, we propose a dual-branch network based on an improved autoencoder. We design a GRD feature extraction module specifically for edge extraction, which effectively captures details and structural information at various scales. Additionally, we incorporate a polarization loss function to further preserve the polarization constraint relationships and prevent their disruption.
- We construct a simulation dataset based on the underwater polarization imaging model and a camera response function (CRF). To verify the generalization performance of the algorithm, we build an underwater polarimetric imaging system and create a real-world dataset. Extensive experimental results on real-world datasets demonstrate the effectiveness of our proposed approach and its superiority against other state-of-the-art methods.

## 2. Related Work

### 2.1. Low-Light Intensity Image Enhancement

Low-light image enhancement methods can be classified into traditional and deep learning-based methods. Traditional methods often utilize techniques such as histogram equalization (HE) [2], Gamma Correction (GC) [12] and the Retinex theory [13]. Histogram equalization methods transform the low-light image enhancement task as a numerical optimization problem, which may lead to inadequate or excessive enhancement and introduce intolerable artifacts, because this method does not consider the local luminance transformation and only considers the global adjustment. Although Gamma Correction can perform non-linear processing on each pixel independently, this method does not consider the relationship between the pixel and its neighboring pixels, which can lead to a decrease in smoothness. Based on the Retinex method, it is assumed that the image has no noise or color distortion, so it usually introduces severe noise or color distortion and requires manual adjustment of parameters.

In recent years, deep learning methods have demonstrated remarkable performance in the field of image processing and have also been applied to low-light image enhancement. These methods can be divided into two categories. The first category of methods uses Convolutional Neural Networks (CNNs) to learn the map from a large amount of data from low-light images to normal images. In 2017, LLNet [4] was proposed as the first deep learning-based method for low-light image enhancement. It employs a variant of the stacked sparse denoising autoencoder to simultaneously enhance brightness and denoise low-light images. In 2018, MELLEN [14] introduced an end-to-end multi-branch enhancement network, incorporating a feature extraction module, an enhancement module, and a fusion module to achieve low-light image enhancement. To alleviate computational burdens, in 2018, Li et al. [15] proposed a lightweight model called LightenNet for low-light image enhancement. Most supervised learning methods are trained on synthetic data and exhibit poor generalization performance in real low-light conditions. In 2021, DSLR [16] was proposed to take advantage of Laplacian pyramids in extracting multi-scale features from images. This method uses Laplacian pyramids in both image and feature spaces

to adjust global illumination and restore local details, effectively preserving local details and avoiding significant color distortion. In 2022, Liu et al. [9] proposed a lightweight Self-Calibrated Illumination (SCI) learning framework, which constructs a self-calibration module that can quickly and effectively enhance low-light images and demonstrates outstanding effectiveness in object detection and semantic segmentation tasks.

To address the data scarcity issue, an unsupervised learning method called Enlighten-GAN [7] is introduced. EnlightenGAN utilizes an attention-guided U-Net as a generator, employing a global–local discriminator to ensure enhanced results resemble real images under normal lighting conditions. Special loss functions are designed to stabilize training. In addition, unlike methods based on image transformation, an unsupervised deep curve estimation network called Zero-DCE [8] was introduced in 2020. Zero-DCE treats low-light enhancement as a curve estimation task for images, taking low-light images as input and generating high-order curves as output. These curves are used to make pixel-level adjustments to the dynamic range input to obtain high-quality images. Additionally, in 2021, the same authors also introduced a fast and lightweight version called Zero-DCE++ [17]. This curve-based method does not require any paired or unpaired data during training and achieves zero-reference learning through a set of non-reference loss functions. The mapping from images to curves only requires a lightweight network, significantly improving computational efficiency. However, these curve transformation methods do not consider noise influence. In 2023, Wang et al. [18] proposed a method to enhance low-light images using the image fusion strategy. This method can automatically simulate the exposure of each image, calculate the perceptual quality based on the contrast, saturation, and saliency of each image, and automatically select useful pixels for fusion to obtain enhanced results. These methods that rely solely on CNNs to learn mappings lack interpretability and theoretical justification.

The second category of methods combined with Retinex theory usually includes multi-stage training. Firstly, different networks are used to realize the tasks of image decomposition, illumination map adjustment, and reflection map recovery, respectively. Then, they are connected for end-to-end fine-tuning. These kinds of deep learning methods based on Retinex theory have stronger interpretability. In 2018, the first deep learning method combining the Retinex theory, Retinex-Net [19], was proposed. It includes a Decom-Net that decomposes the image into illumination and reflection maps, while the Enhance-Net adjusts the illumination map for low-light enhancement. In 2019, Zhang et al. proposed KinD [6], composed of a layer decomposition network, a reflection restoration network, and an illumination adjustment network, to separately handle illumination maps and reflection maps, enabling the training of paired images captured under different exposure conditions. In 2021, an improved method was proposed based on KinD, named KinD++ [20], which effectively fixes the color distortion problem in the KinD method using the Multi-scale Illumination Attention Module (MSIA). However, these methods usually require training a multi-stage network, and the entire training process is cumbersome and time-consuming. In 2019, Wang et al. proposed DeepUPE [21], a single-stage method based on the Retinex theory, which can directly predict the illumination map, avoiding the cumbersome training of a multi-stage network. However, this method did not consider the interference introduced during low-light enhancement, resulting in amplified noise during the enhancement process and introducing additional distortion. In 2021, Liu et al. [22] proposed RUAS (Retinex-inspired Unrolling with Architecture Search), a lightweight framework based on the Retinex theory for optimization model establishment, using a reference-free learning strategy to discover architectures for illumination estimation and noise removal from a compact search space, which can be applied to different types of low-light scenes. In 2023, to suppress noise introduced during the enhancement, Wang et al. [23] proposed a method based on the Retinex theory. The illumination map enhancement module in this method is composed of a new feature extraction structure using depthwise separable convolution, which reduces the size of the network parameters while compensating for some of the shortcomings of poor brightness in dark areas. It better adjusts the brightness

of the illumination map. The reflection map restore module, because it incorporates the illumination map as a reference, makes the texture details of the enhanced image clearer, suppressing noise interference. In addition, these CNN-based methods have limitations in capturing image contextual relevance and non-local self-similarity, which are crucial for image restoration. The recent emergence of the transformer may provide insights for addressing this drawback. In 2023, Cai et al. first proposed a low-light image enhancement algorithm based on the transformer called Retinexformer [24]. This method designs a single-stage low-light image enhancement framework, with a simple training process that can reuse contextual relationships.

While these methods exhibit excellent enhancement performance in numerous applications within the intensity domain, they are not suitable for the enhancement of multiple polarization images. This is because their inputs are individual images, and the networks do not need to consider the physical constraint relationships between multiple images. We attempted to adapt these methods to multiple polarized images, but the enhanced results introduced intolerable artifacts. Polarized image enhancement requires consideration not only of the brightness adjustment but also of the relationships between multiple images.

### 2.2. Low-Light Polarized Image Enhancement

To address the challenges in enhancing low-light polarized images, Hu et al. proposed a dedicated network called IPLNet in 2020 [10]. The network consists of two sub-networks, RGB-Net and Polar-Net. RGB-Net generates three different channels, and Polar-Net predicts polarization information. Additionally, a polarization loss function is designed to balance the network's intensity and polarization features. The effectiveness of the method is validated on indoor and outdoor datasets. While it can enhance multiple low-light polarized images simultaneously, it remains limited to processing polarization information in the intensity domain. In 2023, a study [11] analyzed the average error rate of the image radiance reduction factor for polarization-related variables (including multi-angle polarization images, AoP, DoLP, and Stokes parameters) in the polarization low-light imaging model. The study concluded that, in low-light conditions, the degradation of Stokes parameters is less than that of multiple polarization intensity measurements. Therefore, a new pipeline is proposed to address low-light polarized image enhancement in the Stokes domain rather than the intensity domain. However, this method introduces additional noise when directly amplifying brightness and does not fully leverage the characteristics of different Stokes parameters. Additionally, due to the low number of photons, image quality is affected not only by reduced brightness but also by severe noise. Multiple polarization intensity measurements obtained in low-light conditions contain noise, and polarization parameters are derived from these measurements through a non-linear operator, further amplifying the noise. Therefore, enhancing low-light polarized images requires specific consideration of denoising. A method proposed in [25] is based on Principal Component Analysis (PCA) for denoising, effectively utilizing spatial correlations among various polarization states. Key processes involve dimensionality reduction and linear minimum mean square error estimation in the transformed domain. Another novel denoising algorithm based on K-Singular Value Decomposition (K-SVD) is presented in [26], effectively eliminating Gaussian noise while preserving details and edges. In [27], a polarized image denoising method based on BM3D is introduced, showing noticeable noise removal effects and good robustness.

The non-data-driven methods have some limitations. Firstly, they assume noise to be additive Gaussian white noise, while real noise situations are more complex, leading to poor generalization performance on real data. Secondly, most methods rely on manually set prior assumptions, requiring subsequent manual parameter adjustments. In 2020, a denoising method called PDRDN, based on a residual dense network, was proposed [28]. This network considers multi-channel polarized image inputs and outputs corresponding residual images. PDRDN outperforms traditional methods and represents the first publicly available work on polarized image denoising based on deep neural networks. In 2022, a CNN-based method utilizing an attention mechanism for polarized image denoising was

introduced [29]. The attention mechanism adjusts the contributions of different channels to extract polarization features, proving effective for training multi-channel multi-branch networks. Another contribution of this work is an adaptive polarization loss, enabling the network to focus more on learning polarization information.

## 3. Methods

### 3.1. Underwater Polarization Imaging Model

3.1.1. Traditional Underwater Image Model

The atmospheric scattering model is a classical dehazing model commonly used in image dehazing [30] and some underwater image enhancement methods, as shown in Figure 2. The irradiance $I(x,y)$ received by the detector comes from the superposition of the following two parts.

$$I(x,y) = D(x,y) + B(x,y). \tag{1}$$

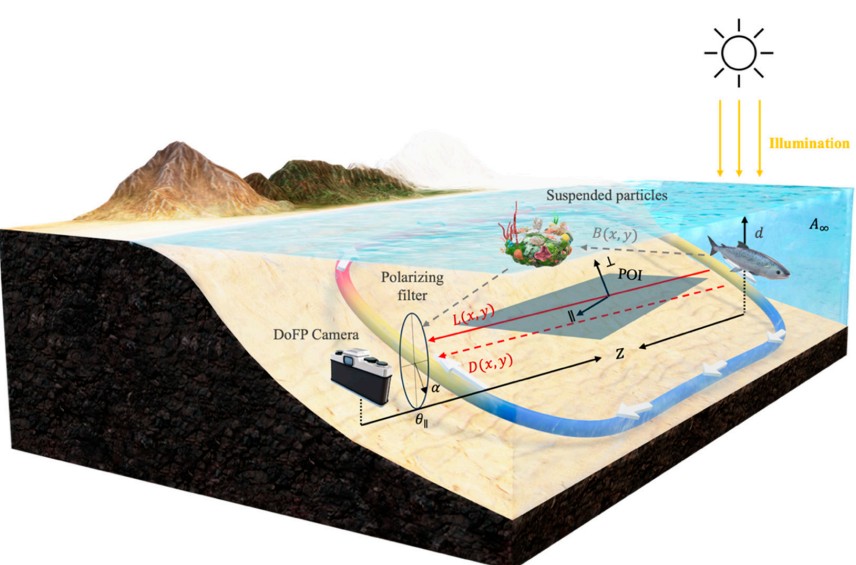

**Figure 2.** Underwater polarization imaging model.

The first source is the signal of an object, whose radiation is attenuated by absorption and scattering in the water, generally referred to as the direct transmission $D(x,y)$, which can be described as the following:

$$D(x,y) = L(x,y)t(x,y) \tag{2}$$

where, $(x,y)$ represents the coordinates of the pixels in the image, $L(x,y)$ represents the original object radiance before attenuated on the line of sight (LOS), and $t(x,y)$ represents the transmittance of the medium, which can be expressed as follows:

$$t(x,y) = e^{-\beta z} \tag{3}$$

where, $z$ represents the shooting distance, and $\beta = \{\beta(c)\}$ is the attenuation coefficient, which is related to the absorption and scattering effects of the water body.

The second source comes from the light that part of the ambient light is scattered towards the camera by particles in the water, which is generally called veiling light or backscatter signal $B(x,y)$, which can be expressed as:

$$B(x,y) = A_\infty[1 - t(x,y)] \tag{4}$$

Among them, $A_\infty$ indicates the ambient light where the LOS extends to infinity, which is generally related to the particles in the water and the lighting conditions. The above model based on atmospheric imaging characteristics considers $t(x,y)$ to be the same in

forward scattering and backward scattering, which is in line with the negligible absorption of light in the atmosphere, and can achieve better results in air dehazing, but in underwater turbid environment, the absorption and scattering of light cannot be ignored. To better simulate the underwater environment, we use the modified underwater imaging model in the literature [31], in which $t(x, y)$ in forward scatter is replaced with $T_c^D$, and $t(x, y)$ in backscatter is replaced with $T_c^B$.

$$T_c^D = e^{-\beta_c^D(v_D)z} \tag{5}$$

$$T_c^B = e^{-\beta_c^B(v_B)z} \tag{6}$$

where, $\beta_c^D$ is the direct transmission attenuation coefficient, $\beta_c^B$ is the backscatter transmission attenuation coefficient. $v_B = \{E, S_c, b, \beta\}$ is affected by ambient light $E$, camera response $S_c$, scene reflectivity $\rho$, water scattering coefficient $b$ and water attenuation coefficient $\beta$.

### 3.1.2. Underwater Polarization Imaging Model

Given a series of multiple polarization intensity measurements, i.e., $I_{\alpha \in \{1,2,3,4\}}$, the Stokes vector notation can quantitatively describe the polarization and intensity information with four Stokes parameters. Our approach accomplishes image enhancement by modulating the linear polarization component, whereas unlike circularly polarization light, which has four Stokes parameters, linearly polarization light can be fully represented by three parameters $S_0, S_1, S_2$. The formula is as follows:

$$\begin{aligned} S_0 &= \tfrac{1}{2}(I_1 + I_2 + I_3 + I_4) \\ S_1 &= I_3 - I_1 \\ S_2 &= I_4 - I_2 \end{aligned} \tag{7}$$

where the parameter $S_0$ is the intensity image, which describes the total intensity of the light field. The parameters $S_1$ and $S_2$ represent the difference in linearly polarization light. DoLP describing the polarization characteristics can be calculated as.

$$\text{DoLP} = \frac{\sqrt{S_1^2 + S_2^2}}{S_0} \tag{8}$$

The pixels of DoLP are constant between 0 and 1, indicating the proportion of polarization light in the total light intensity. Figure 3 shows a diagram of the multiple polarization intensity measurements, the Stokes parameter, and the polarization image.

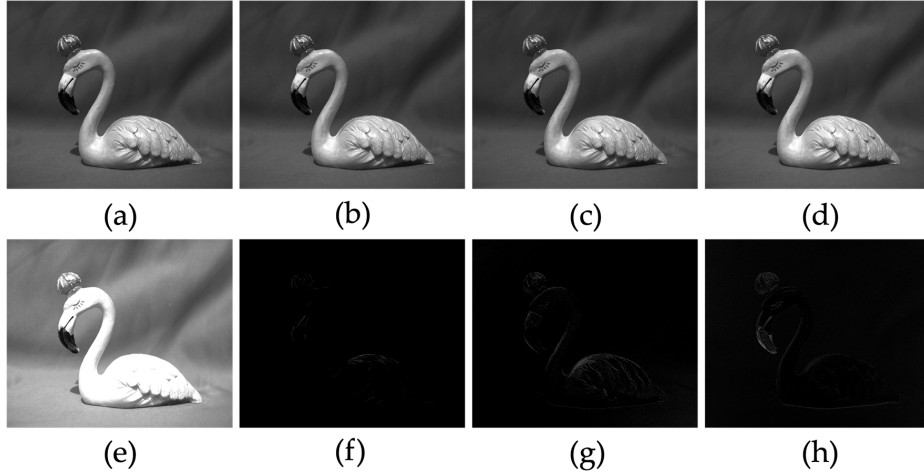

**Figure 3.** Visualization of methods for describing polarization information: (**a**–**d**) images from the dataset [32] with different polarization angles ($I_{0°}$, $I_{45°}$, $I_{90°}$, $I_{135°}$); (**e**–**g**) corresponding Stokes parameters ($S_0$, $S_1$, $S_2$); and (**h**) DoLP image.

Next, we will discuss the influence of polarization effects in underwater imaging models. It is worth noting that both forward scatter and backscatter contribute to polarization. Some traditional polarization imaging methods ignore the contribution of forward scattering, which will lead to a decrease in performance. First, we assume that the illumination of any scattered particle comes from one direction, and the parallel lines emitted from the camera to the scattered particles are defined as a plane of incidence (PoI), then there are two directions, parallel and perpendicular, and these two directions represent the two images with the greatest difference according to the idea of polarization differential imaging. Then, our above imaging model can be divided into two directions: $I^{\parallel}$ and $I^{\perp}$. According to Equation (1), $I^{\parallel}$ and $I^{\perp}$ can be expressed as:

$$I^{\parallel}(x,y) = D^{\parallel}(x,y) + B^{\parallel}(x,y) \tag{9}$$

$$I^{\perp}(x,y) = D^{\perp}(x,y) + B^{\perp}(x,y). \tag{10}$$

Similarly, the total intensity of the obtained irradiance can be expressed as:

$$I(x,y) = I^{\parallel}(x,y) + I^{\perp}(x,y). \tag{11}$$

Then, $D$ and $B$ in the imaging model can also be represented as:

$$D(x,y) = D^{\parallel}(x,y) + D^{\perp}(x,y) \tag{12}$$

$$B(x,y) = B^{\parallel}(x,y) + B^{\perp}(x,y). \tag{13}$$

Similarly, DoLP, denoted as $P$ in the formula, can be expressed as follows (we assume that $I^{\parallel}(x,y) \geq I^{\perp}(x,y)$):

$$P(x,y) = \frac{I^{\parallel}(x,y) - I^{\perp}(x,y)}{I^{\parallel}(x,y) + I^{\perp}(x,y)} = \frac{I^{\parallel}(x,y) - I^{\perp}(x,y)}{I} \tag{14}$$

$$P_D(x,y) = \frac{D^{\parallel}(x,y) - D^{\perp}(x,y)}{D^{\parallel}(x,y) + D^{\perp}(x,y)} = \frac{D^{\parallel}(x,y) - D^{\perp}(x,y)}{D} \tag{15}$$

$$P_B(x,y) = \frac{B^{\parallel}(x,y) - B^{\perp}(x,y)}{B^{\parallel}(x,y) + B^{\perp}(x,y)} = \frac{B^{\parallel}(x,y) - B^{\perp}(x,y)}{B} \tag{16}$$

Since the scattered light is partially polarized in the perpendicular direction of the PoI, $P$, $P_D$ and $P_B$ are not less than zero. In [33], it is pointed out that the polarization properties $P$ of transmitted light depend on the material properties of the scene objects (e.g., surface textures), and that objects in the same semantic segment tend to have similar material properties. Thus, $P$ is spatially varied, while $P_D$ and $P_B$ are in the same semantic segment, and $P_D$ is approximately uniform, while $P_B$ can be seen as spatially homogeneous, and the $P_D$ can be simulated according to the semantic information S.

To calculate the polarization information, it is necessary to obtain multiple polarization intensity measurements; the principle is to place a polarizer in front of the camera, when the polarization angle is $\alpha$, according to Marius's law, the captured polarization image $I_\alpha$ can be calculated as:

$$I_\alpha = \frac{I \cdot \left(1 - P \cdot cos\left(2\left(\alpha - \theta_{\parallel}\right)\right)\right)}{2} \tag{17}$$

where, $\theta_{\parallel} = \left\{ \overline{\theta_{\parallel}} + N(x,y,c) \right\}$ denotes the direction of the polarizer, i.e., the optimal transmission components of the parallel PoI. $D_{\alpha}$ and $B_{\alpha}$ also have:

$$D_{\alpha} = \frac{D \cdot \left( 1 - P_D \cdot cos\left( 2\left( \alpha - \theta_{\parallel} \right) \right) \right)}{2} \tag{18}$$

$$B_{\alpha} = \frac{B \cdot \left( 1 - P_B \cdot cos\left( 2\left( \alpha - \theta_{\parallel} \right) \right) \right)}{2} \tag{19}$$

From the above equation, it can be deduced:

$$I \cdot P = D \cdot P_D + B \cdot P_B \tag{20}$$

which means that $D$ and $B$ are determined by $P$, $P_D$, $P_B$. According to Equation (21):

$$P = \frac{D \cdot P_D + B \cdot P_B}{I} \tag{21}$$

$P$ can be easily calculated. Once $P$ is obtained, we can calculate the $I_{\alpha}$ of the polarized image according to Marius's law.

### 3.2. Polarization-Based Low-Light Image Enhancement Pipeline

Most of the existing underwater low-light image enhancement methods are processed in the intensity domain, which only rely on the intensity information, and the introduction of polarization information can improve the imaging performance in low-light scenes. There are two main problems in the existing deep learning intensity domain methods to deal with polarization images, one is that the noise distribution of polarization images is more complex than intensity images, and directly amplified brightness may introduce new noise influences, and the independently designed denoising module in the network often assumes that the noise is Gaussian white noise, which is not good for complex polarization images. Second, the input of the network is often an individual image in a scene, relying on the network to learn the feature distribution of the image, which does not apply to multiple polarization images in the same scene, and the independent processing of multiple images by the network will destroy the polarization constraint relationship and introduce additional interference and noise (this can be verified from the experimental results in Section 4.3.1). Therefore, the underwater low-light polarization image enhancement method needs to study how to better combine the network design to extract polarization information and achieve high-quality underwater low-light image enhancement based on not destroying the polarization relationship. Our goal is to obtain high-quality DoLP. Inspired by [11], the degree of degradation in the Stokes domain is smaller than that intensity domain under low-light conditions, so we can solve the problem in the Stokes domain instead of the intensity domain.

Using the network to learn the mapping between Stokes parameters independently will destroy the polarization constraint relationship, and we hope to further explore how to enhance the Stokes parameters in the same scenario without breaking the polarization relationship. Through a large number of experimental observations, we have stumbled upon different characteristics of the physical properties of the Stokes parameter. For the low-light enhancement task, after directly magnifying the brightness by 10 times, as shown in Figure 4, $S_0$ is less affected by noise, and although most areas in $S_1$ and $S_2$ are degraded by noise, their edges are less affected, which can provide rich detailed structural information. This is due to the differential antagonistic mechanism calculated $S_1$ and $S_2$.

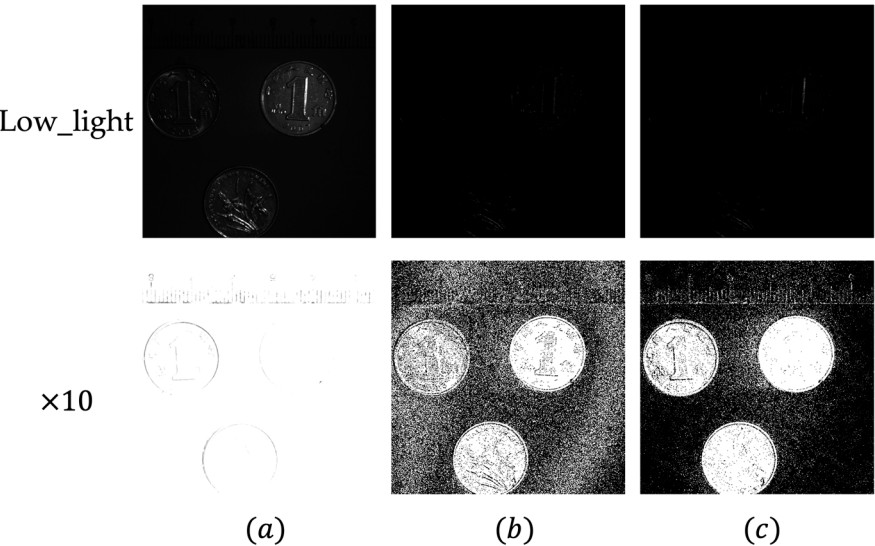

Low_light

×10

(a)    (b)    (c)

**Figure 4.** (**a**) $S_0$ is not sensitive to noise, and the degradation is similar to an intensity image, (**b**,**c**) $S_1$ and $S_2$ have an antagonistic relationship.

Inspired by the observation, we propose a Stokes-domain underwater low-light polarized image enhancement paradigm based on the physical characteristics of Stokes parameters. The Stokes parameters $S_{i \in \{0,1,2\}}$ are obtained from the multiple polarization intensity measurements $I_{\alpha \in \{1,2,3,4\}}$ utilized in Equation (7). We aim to enhance $S_{i \in \{0,1,2\}}$ to obtain high-quality DoLP. Two independent branch networks are used to enhance $S_0$ and $S_{1,2}$ separately. The Brightness Adjustment Network is designed for $S_0$ focusing on improving brightness, while the Detail Enhancement Network inputs $S_1$ and $S_2$ to focus on enhancing edge details by using their antagonistic relationship. Then, the $curve(\cdot)$ obtained by the Brightness Adjustment Network is used to adjust the brightness of the enhanced $\hat{S}_{1,2l}$, and this adaptive adjustment of brightness effectively reduces the noise introduced by directly amplifying the brightness. The polarization loss $L_{polar}$ is further utilized to constrain the maintenance of polarization relationships. The overall framework of our polarization-aware low-light image enhancement algorithm is illustrated in Figure 5.

$$\hat{S_{0h}} = f_{BA}(S_{0l}), \; curve = \hat{S_{0h}} - S_{0l} \tag{22}$$

$$\hat{S_{1,2l}} = f_{DE}(S_{1,2l}) \tag{23}$$

$$S_{1,2h} = curve(\hat{S_{1,2l}}). \tag{24}$$

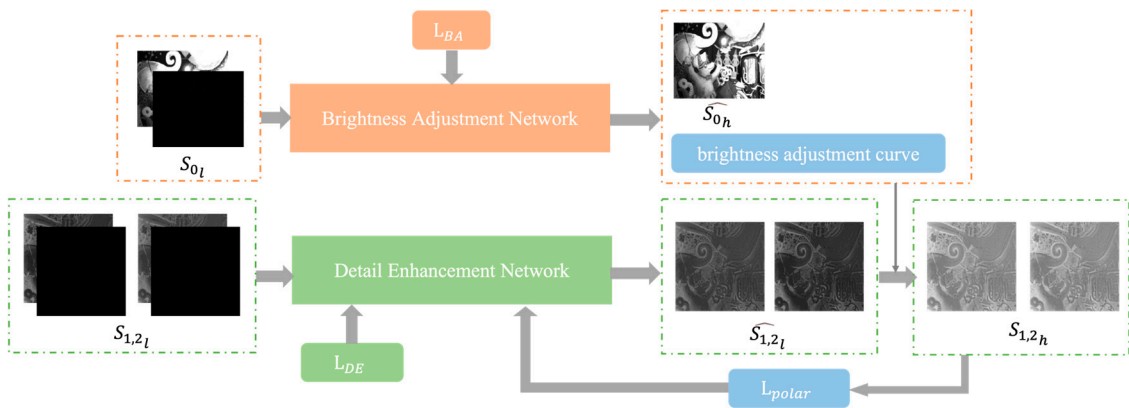

**Figure 5.** The overall framework of the proposed polarization-aware low-light underwater image enhancement algorithm.

### 3.3. Polarization-Based Low-Light Image Enhancement Network

#### 3.3.1. Brightness Adjustment Network

The Brightness Adjustment Network based on the Retinex is designed. The Retinex theory posits that the observed image is composed of an illumination map $I(x, y)$ and a reflectance map $R(x, y)$. $I(x, y)$ capture ambient lighting information, and $R(x, y)$ describe the intrinsic properties of the target, which should remain consistent under any brightness conditions. The Brightness Adjustment Network comprises the Layer Decomposition Module (LD), the Reflectance Restoration Module (RR), and the Illumination Adjustment Module (IA). Specifically, LD is responsible for decomposing the input $S_0$ into a reflectance map $R(x, y)$ and an illumination map $I(x, y)$, RR addresses degradation issues, and IA adjusts brightness. The design of the Brightness Adjustment Network is inspired by [19] and the network structure is shown in Figure 6.

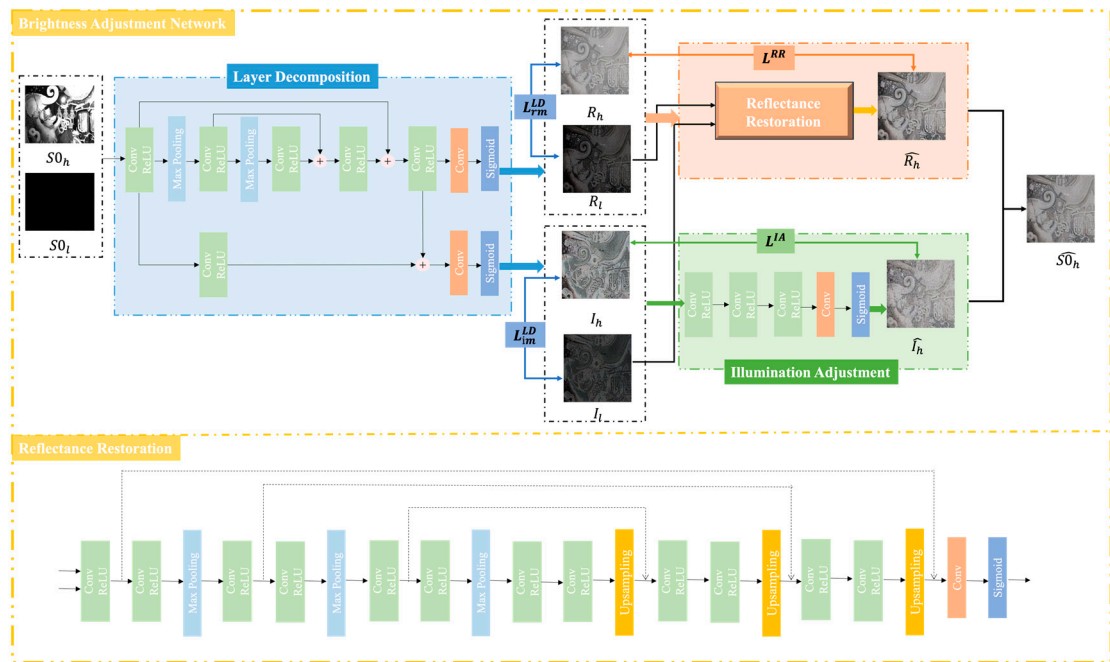

**Figure 6.** The architecture of Brightness Adjustment Network.

**Layer Decomposition Net.** The layer decomposition network consists of two branches, corresponding to the reflectance and illumination map, respectively. We divide the dataset into $S_{0l}$ and $S_{0h}$ based on the exposure information, representing low-light and high-light images, respectively. The decomposed reflectance map and illumination map are represented as $[R_l, R_h]$ and $[I_l, I_h]$. Recovering two components from an image is a highly undetermined problem, so we designed three loss functions to constrain the similarity of $R_i$, $I_i$ and $S_{0i}$. $L_{rm}^{LD}$ represents the loss of reflection component consistency, to constrain the consistency of the reflection component $R_i$ pairs of images.

$$L_{rm}^{LD} = \|R_l - R_h\|_2^2 \tag{25}$$

$L_{im}^{LD}$ is the smoothing loss of the lighting component, which is based on the assumption of the lighting component; ideal lighting $I_i$ component should be as smooth as possible in texture detail, and the loss function is expressed as assigning weights to the gradient map of the $I_i$ by finding the gradient $R_i$ of the reflection component, and using the reflection component to constrain the smoothed area of the $R_i$ in the smoothing area.

$$L_{im}^{LD} = \sum_{i=l,h} \|\nabla I_i \times exp(-\lambda_g \nabla R_i)\| \tag{26}$$

$L_{rec}^{LD}$ is the reconstruction loss, and the decomposed image should be able to be re-assembled into the pre-decomposition image.

$$L_{rec}^{LD} = \|S_{0l} - R_l \times I_l\| + \|S_{0h} - R_h \times I_h\|. \tag{27}$$

Therefore, the total loss function of the layer decomposition network is as follows, and $\lambda_1$, $\lambda_2$ and $\lambda_3$ are, respectively, set to 1, 0.01, 0.08.

$$L^{LD} = \lambda_1 L_{rec}^{LD} + \lambda_2 L_{rm}^{LD} + \lambda_3 L_{im}^{LD}. \tag{28}$$

**Reflectance Restoration Net.** The network structure is a deeper U-Net structure. Since we focus more on brightness enhancement, the loss function for RR is relatively simple, where $\hat{R}$ represents the reconstructed reflectance maps.

$$L^{RR} = \left\|\hat{R}_h - R_h\right\|_2^2. \tag{29}$$

**Illumination Adjustment Net.** The network consists of three Conv + ReLU layers, followed by one Conv layer with $3 \times 3$ kernel size and a Sigmoid layer. The loss function is as follows, where $\hat{I}_h$ is the reconstructed illumination maps.

$$L^{IA} = \left\|\hat{I}_h - I_h\right\|_2^2 + \left\|\left|\nabla \hat{I}_h\right| - \left|\nabla I_h\right|\right\|_2^2. \tag{30}$$

Finally, we use the $\hat{S_{0h}}$ and the $S_{0l}$ to calculate a *curve(·)*, which will be used to adjust the $\hat{S_{1,2l}}$ obtained by the Detail Enhancement Network.

### 3.3.2. Detail Enhancement Network

Although $S_{1,2}$ is seriously affected by noise, it still retains a relatively complete target edge, which can provide rich structural information. We design a gradient residual dense feature extraction module (GRD) based on the attention mechanism, including feature extraction blocks composed of convolutional layers to extract the features of the input image $S_{1,2}$ features and gradient residuals dense blocks based on the attention mechanism to extract the edge information of $S_{1,2}$, which can extract deep features with rich fine-grained detail information, and then input into the backbone to complete image reconstruction, the backbone is represented by $E$, and the whole process can be described as follows:

$$\hat{S}_i = f_{DE}(S_i) = E(conv(S_i) \oplus dense(\nabla S_i)) + S_i. \tag{31}$$

To handle multi-scale inputs, we use two downsampling and upsampling blocks, embedding three Denseblocks [34] in the middle to obtain more comprehensive contextual information, and adding skip-connections to maximize the use of shallow features. The downsampling block adopts a residual bottleneck structure [35] and introduces a channel shuffle operation [36] to enhance the information flow. The design of the upsampling block is inspired by Attention U-Net [37] and is used to efficiently merge high-resolution and low-resolution features. Figure 7 shows the network structure of the Detail Enhancement Network.

The Detail Enhancement Network focuses on the structural similarity between the ground truth and the output of the network, we choose the smoothness of the total variational loss $L_{tv}$ to improve the results and the gradient loss $L_{grad}$ to keep the structure invariant and add $L_1$ loss and $L_2$ loss to improve the stability and generalization ability of network training.

$$L_1 = \left\|\hat{S_1} - S_1\right\|_1 + \left\|\hat{S_2} - S_2\right\|_1 \tag{32}$$

$$L_2 = \left\|\hat{S_1} - S_1\right\|_2 + \left\|\hat{S_2} - S_2\right\|_2 \tag{33}$$

$$L_{tv}(x) = \frac{1}{N}\sum_i^N \left(\sum_{h,w,c} (x_{h+1,w,c} - x_{h,w,c})^2 + (x_{h,w+1,c} - x_{h,w,c})^2\right) \tag{34}$$

$$L_{grad} = \sum\nolimits_{i=1,2} MSE\left(||\nabla(\hat{S}_i)|, |\nabla(S_i)||\right) \tag{35}$$

where $x$ represents the image data, which in this branch is $S_{1,2}$, and its shape is $N \times c \times h \times w$, $n$ is the batch size, $c$ is the number of channels, and $h$ and $w$ are the height and width of the image. In addition, to better reconstruct the polarization information, we have also introduced the polarization loss $L_{polar}$:

$$L_{polar} = \left\|D\hat{o}LP - DoLP\right\|_2^2. \tag{36}$$

The total loss function is defined as follows, where $\lambda_1$, $\lambda_2$, $\lambda_3$, $\lambda_4$ and $\lambda_5$ are, respectively, set to 10, 100, 1, 100 and 20.

$$L_{DE} = \lambda_1 \cdot L_1 + \lambda_2 \cdot L_2 + \lambda_3 \cdot L_{tv} + \lambda_4 \cdot L_{grad} + \lambda_5 \cdot L_{polar}. \tag{37}$$

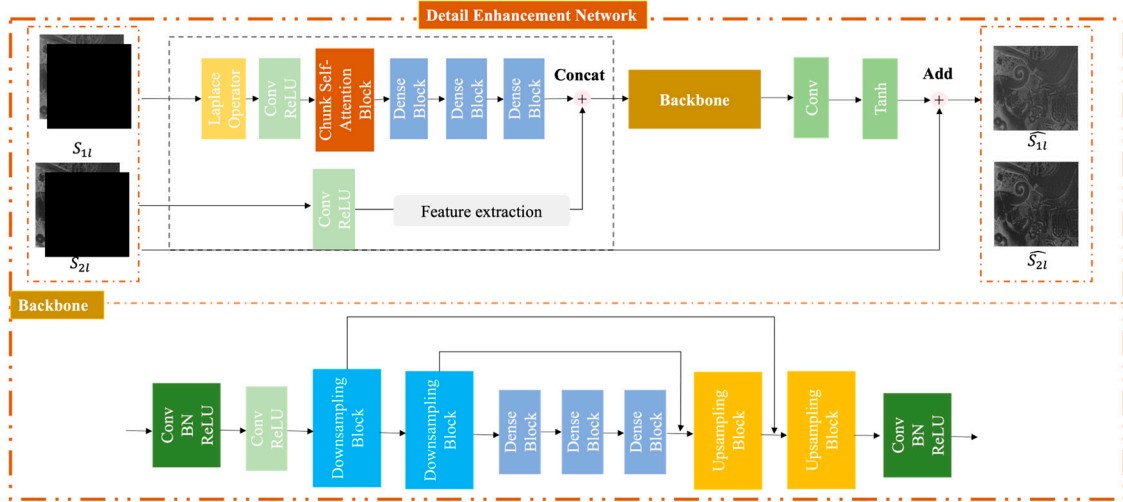

**Figure 7.** The architecture of Detail Enhancement Network.

## 4. Experiments

### 4.1. Data Preparation

Currently, there is no publicly available underwater polarization image dataset. The existing low-light image enhancement dataset only contains intensity information and cannot be used to generate polarized images, and the existing polarized image dataset [32] has only a single target in the scene, and the scene depth information is simple, as shown in Figure 3, making it difficult to simulate the real underwater scene. Therefore, we construct an underwater low-light polarization image dataset, including the real-world acquired by polarization camera and simulated polarization dataset, we train the network with the simulated data, test the performance on the real-world dataset, and compare it with other advanced methods.

#### 4.1.1. Simulation Dataset

According to the underwater polarization imaging model introduced in Section 3.1.1 above, we use the scene depth map $z$ to restore the real underwater scene as much as possible. $\beta_c^D$ and $\beta_c^B$ can be obtained using the following formula:

$$\beta_c^D = ln\left[\frac{\int_{\lambda_1}^{\lambda_2} S_c(\lambda)\rho(\lambda)E(d,\lambda)e^{-\beta(\lambda)z}d\lambda}{\int_{\lambda_1}^{\lambda_2} S_c(\lambda)\rho(\lambda)E(d,\lambda)e^{-\beta(\lambda)(z+\Delta z)}d\lambda}\right]/\Delta z \tag{38}$$

$$\beta_c^B = -ln\left[1 - \frac{\int_{\lambda_1}^{\lambda_2} S_c(\lambda) B^\infty(\lambda)\left(1 - e^{-\beta(\lambda)z}\right)d\lambda}{\int_{\lambda_1}^{\lambda_2} S_c(\lambda) B^\infty(\lambda)d\lambda}\right]/z. \tag{39}$$

The ambient light $E$ and $B^\infty$ are related to the superficial and optical properties (AOPs) and intrinsic optical properties (IOPs) of water, and the data of AOPs and IOPs can be obtained from related studies [38], and the camera response $S_c(\lambda)$ can also be obtained in [39], in which we set $\rho$ to 1, and use CIE D65 as the surface light.

For the simulation of polarization information, through the analysis in Section 3.1.2 above, we need semantic segmentation labels $S$ to generate reasonable $P_D$ and $P_B$. Therefore, our raw dataset needs to meet the following two conditions:

(1)    Clear image $L(x, y)$ with depth map $z$. Using Equation (1), we can generate $I(x, y)$ by simulating $\beta_c^D$, $\beta_c^B$ and $A_\infty$. $\beta_c^D$, $\beta_c^B$ settings refer to [31]. $A_\infty$ is in [0.85, 0.95].
(2)    Paired semantic segmentation labels $S$. We can generate reasonable $P_D$ based on semantic information, generate $P_D$ from $S$. $P_D = f(S)$ (in [0.025, 0.2]), $P_B$ (in [0.05, 0.4]).

Cityscapes-DBF [40] is widely used in scene understanding and semantic segmentation tasks. It is composed of high-resolution images of multiple cities in Germany, the dataset covers a variety of weather conditions and urban environments, and each image is divided into labeled and unlabeled parts, and the labeled part provides pixel-level semantic segmentation labels, covering categories such as roads, buildings, vehicles, pedestrians, etc. It is convenient for us to use semantic information to assign similar polarization properties to objects in the same semantic segment.

Most of the low-light image simulation methods simulate the noise impact of the real low-light environment by adjusting the brightness of the image and then adding noise. To get as close to the real world as possible and not break the polarization constraint relationship, we use the polarization camera response function (CRF) to simulate the low-light images. Since the images acquired by the polarization camera have different polarization directions, it is necessary to acquire CRF separately for each polarization direction. Polarization images taken from four angles are used as baseline images, and CRF based on luminance values are obtained from these baseline images. The synthesis process is shown in Figure 8.

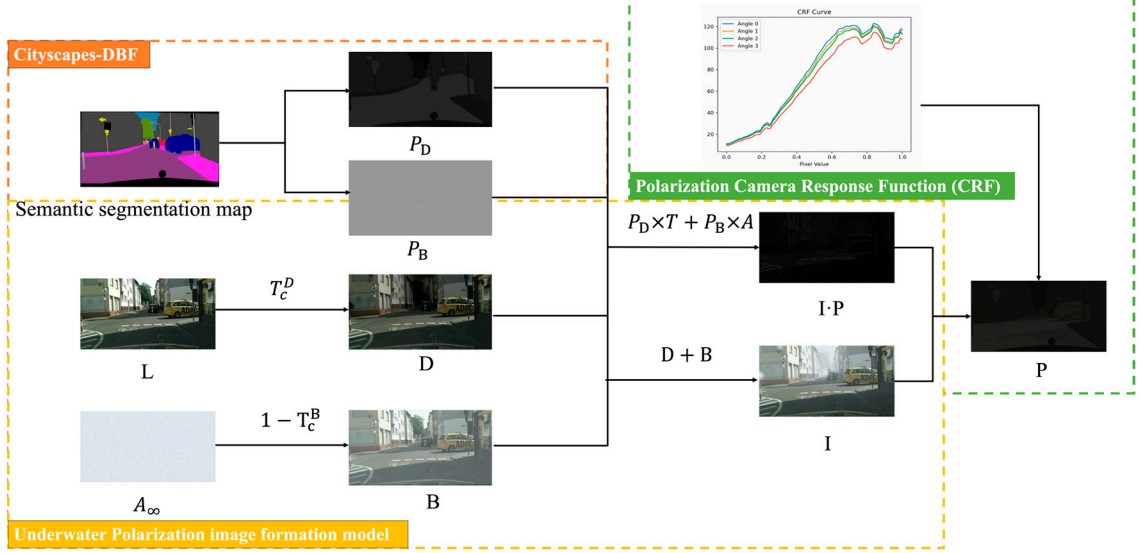

**Figure 8.** Simulation dataset synthesis process.

We select 2975 images to simulate, and the final dataset is expanded to 14,875 groups by cropping, each of which includes a four-angle polarization image in low light and normal light. One group of the simulation dataset is shown in Figure 9.

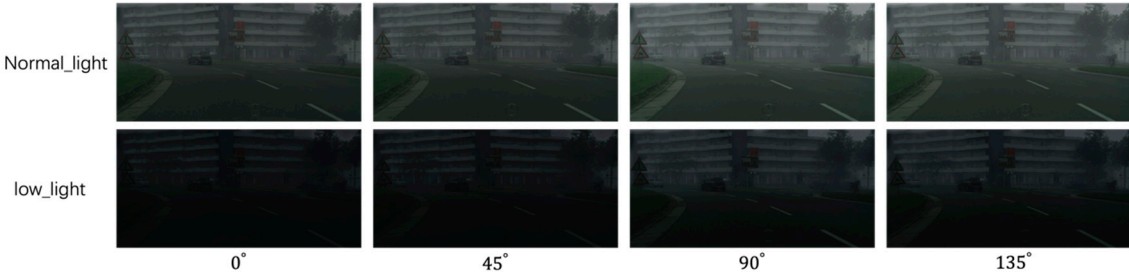

**Figure 9.** Example of simulation dataset.

### 4.1.2. Real-World Datasets

To verify the generality of the algorithm, we set up an underwater polarimetric imaging system. In the experiment, the shooting target is placed in a water tank with a size of $50 \times 27 \times 30$ cm$^3$, and we use a Divide of Focal Plane (DoFP) Hikvision monochrome polarization camera (MV-CH050-10UP) (Hikvision, Hangzhou, China) to acquire polarized images with a spatial resolution of $2048 \times 2448$. The experiment uses active illumination, and the polarization state generator (PSG) generates linearly polarization light through an LED light source with a linear polarizer (extinction ratio 1000:1, diameter 50.8 mm). The experimental setup is shown in Figure 10. Both the polarization camera and the PSG are fixed on a tripod to ensure that the relative position remains unchanged throughout the shooting process.

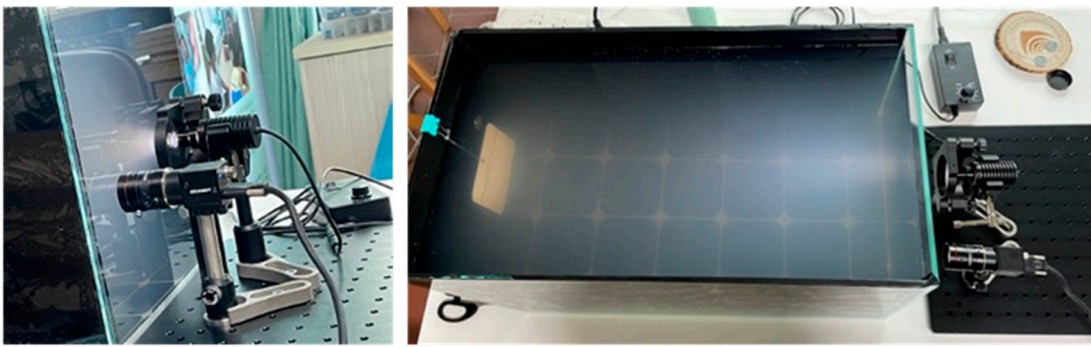

**Figure 10.** Underwater polarimetric imaging system.

The polarization camera can acquire four polarization images of different angles $(I_{0°}, I_{45°}, I_{90°}, I_{135°})$ at one time, which is due to the polarization camera's CMOS sensor integrating four angle polarizers into one sensor. The internal structure of the polarization camera is shown in Figure 11. Shooting four polarization images at different angles at once avoids the error caused by rotating the polarizer.

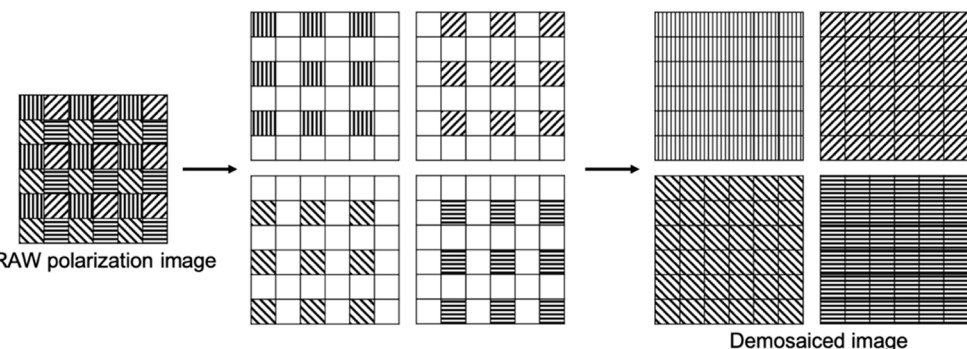

**Figure 11.** The principles of polarization camera imaging.

To facilitate the subsequent experimental evaluation, we change the camera exposure time to make image pairs with pixel alignment. A long exposure time means that the

camera's sensor can receive more photons, and the normal-light image is used as ground truth, while a short exposure time means that the camera's sensor receives fewer photons, resulting in a low-light image. In this way, we can obtain the corresponding ground truth for each low-light image. To reduce the influence of noise, we collect five images for each scene and keep other camera parameters unchanged except for the exposure time for each shot. Then, we calculate the average value as the ground truth. We select 300 groups of real data, and an example of the dataset is shown in Figure 12.

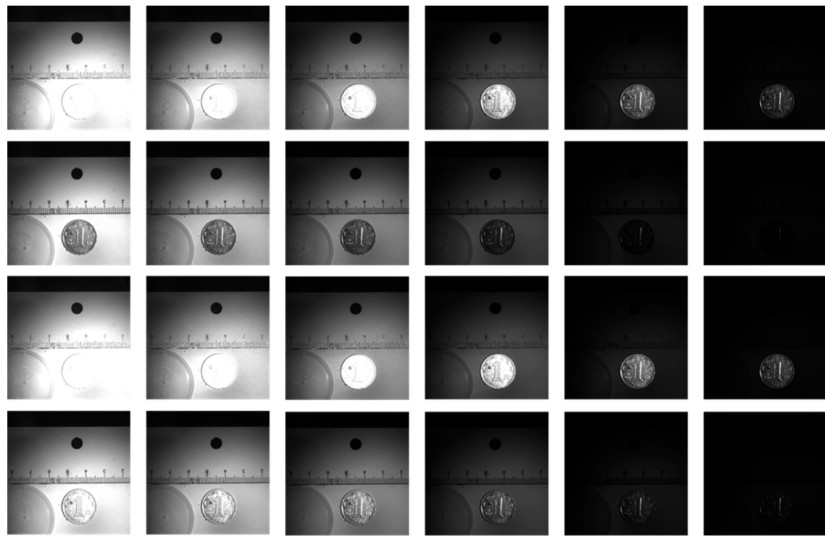

**Figure 12.** Real-world underwater polarization datasets with different illumination.

To simulate the real water environment, we analyzed the characteristics of various scattering particles and finally chose to simulate the underwater scattering medium by adding different amounts of milk to clean water. We then recorded images with degenerative polarization information in the scattering environment after uniform diffusion of the milk. Furthermore, to ensure the diversity of the dataset and analyze the influence of the polarization characteristics of different materials, we select different materials objects, including metal, plastic, paper, wood, glass, resin, etc. The metal target's polarization characteristics are the most obvious, and we also arrange targets with different materials in the same scene to verify the proposed method's robustness. Examples of acquired datasets for different targets are shown in Figure 13.

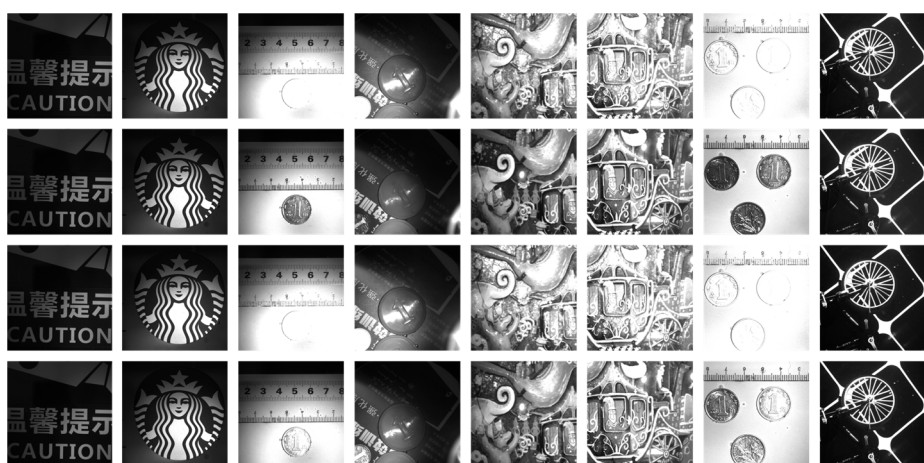

**Figure 13.** Real-world underwater polarization dataset with different material targets.

## 4.2. Training Details

We used the simulation dataset to train the network and selected 32 sets of data with different turbidity and polarization characteristics in real-world underwater polarization dataset as the test dataset. To speed up network training and enrich the dataset, we crop all images in the training dataset to the size of 256 × 256 and expand the cropped data by flipping and rotating. We implement the network using PyTorch and train it for 400 epochs with a batch size of 8. The network optimizer uses the Adam (Adaptive Moment Estimation) optimizer (default parameters $\beta_1$ = 0.9, $\beta_2$ = 0.999 [41]). The Adam optimizer adaptively calculates different learning rates by comprehensively considering the mean and variance of the loss gradient, and limits them to a certain range, so that the update of network parameters is more stable, simple calculation is efficient, and the memory requirement is low. We run all the experiments on an NVIDIA 3090Ti GPU.

## 4.3. Qualitative and Quantitative Analysis

Considering that there are two main problems faced by low-light polarization image enhancement, brightness enhancement and noise removal. Therefore, in this section, we compare our method with the low-light enhancement method and the denoising method on the real-world dataset, respectively.

### 4.3.1. Qualitative Analysis of Final Results

For the denoising method, we chose the traditional denoising method BM3D [27] and the learning-based method Noise2Noise [42]. For both methods, we use two enhancement solutions, one is to denoise the DoLP directly, naming as BM3D_1 and Noise2Noise_1, and the other is to denoise the Stokes parameter and then calculate the DoLP, naming as BM3D_2 and Noise2Noise_2. The experimental results of the denoising method are shown in Figure 14. BM3D and Noise2Noise are both blind denoising methods, and the BM3D method needs to select the noise type, and the results given here are the best results obtained by our artificial parameter tuning. The results of BM3D_1 and Noise2Noise_1 are better, and the results of BM3D_2 and Noise2Noise_2 are very bad, which seriously damages the structural stability and introduces unbearable artifacts. This is because the blind denoising methods will introduce new interference with denoising. The result obtained by the first method is also unable to recover target information, and the denoising effect is not good, because the DoLP's noise obtained in the low-light situation is very complex, a variety of noise models are mixed, and most of the existing denoising methods assume that the noise is following the Gaussian distribution, which is obviously not suitable for the situation of low-light acquisition of polarized images.

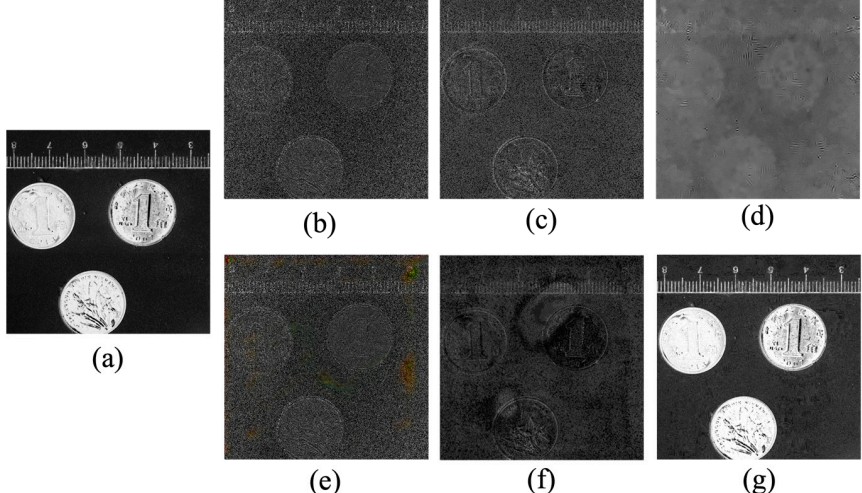

**Figure 14.** Visual comparison on real-world dataset among denoise approaches: (**a**) ground truth; (**b**) input; (**c**) BM3D_1; (**d**) BM3D_2; (**e**) Noise2Noise_1; (**f**) Noise2Noise_2; and (**g**) ours.

For the low-light enhancement method, we select eight intensity domain low-light enhancement methods, including Retinex-Net [19], MBLLEN [14], KinD [6], Enlighten-GAN [7], DSLR [16], ZeroDCE [8], RUAS [22], SCI [9] and a polarization low-light enhancement method Polar [11]. For metal targets with obvious polarization such as coins and rulers, the experimental results are shown in Figure 15. It can be seen from the results that Retinex-Net, MBLLEN, and EnlightenGAN have excessive enhancement, and MBLLEN and DSLR have serious structural distortion. This is due to the lack of guidance from polarization constraints, and the direct enhancement of the $S_1$ and $S_2$ will amplify the noise, which will be further amplified by nonlinear calculations. Although the KinD method remains stable, the overall noise is still very serious. Although the ZeroDCE, RUAS, and SCI methods maintain good structural integrity overall, the effect of observing image detail recovery is not good, because there is no specific detail information extracted for polarization parameters. In summary, the intensity domain method enhances multiple images of the same scene in isolation, resulting in poor structural stability and poor image detail recovery. Polar, the polarization domain method, performs well in structural stability but is slightly inferior to our method in detail recovery and brightness adjustment. Our method achieves the best results in terms of both noise reduction and brightness recovery compared to other methods.

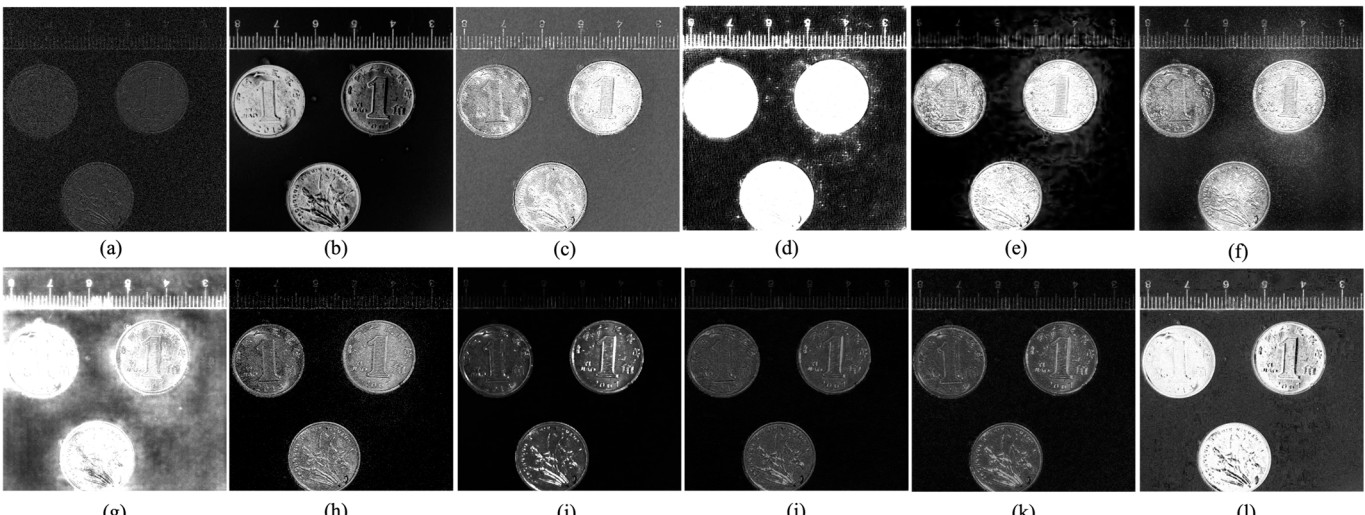

**Figure 15.** Comparison of metal target enhancement results: (**a**) input; (**b**) ground truth; (**c**) Retinex-Net; (**d**) MBLLEN; (**e**) KinD; (**f**) EnlightenGAN; (**g**) DSLR; (**h**) ZeroDCE; (**i**) RUAS; (**j**) SCI; (**k**) Polar; and (**l**) ours.

For targets with different polarization materials in the same scene, such as plastic chips and metal coins, the experimental results are shown in Figure 16. Retinex-Net, MBLLEN, KinD, EnlightenGAN, and DSLR have introduced obvious semicircular artifacts, in which MBLLEN and DSLR both have blurred edges and distorted overall structure, while ZeroDCE, RUAS, SCI methods maintain overall structure, but only have a good recovery effect on metal target coins with obvious polarization, and the "scale" recovery effect of details is not obvious. The polarization enhancement method Polar is not as good as our method for recovering the edge details of the scale and plastic chips, and our method is more balanced for different polarization materials.

### 4.3.2. Qualitative Analysis of Intermediate Results

To further demonstrate the effectiveness of our proposed dual-branch network design, we provide intermediate results for the dual-branch network. For the Brightness Adjustment Network, we compare it with the above low-light enhancement methods. Figure 17 shows the intermediate result of parameter $S_0$.

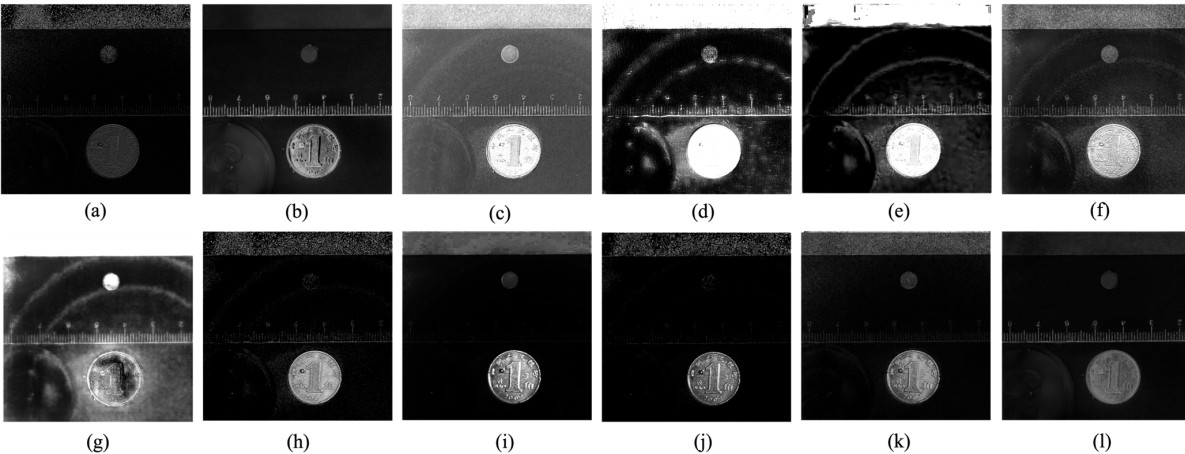

**Figure 16.** Contrast of multiple polarization materials target enhancement results: (**a**) input; (**b**) ground truth; (**c**) Retinex-Net; (**d**) MBLLEN; (**e**) KinD; (**f**) EnlightenGAN; (**g**) DSLR; (**h**) ZeroDCE; (**i**) RUAS; (**j**) SCI; (**k**) Polar; and (**l**) ours.

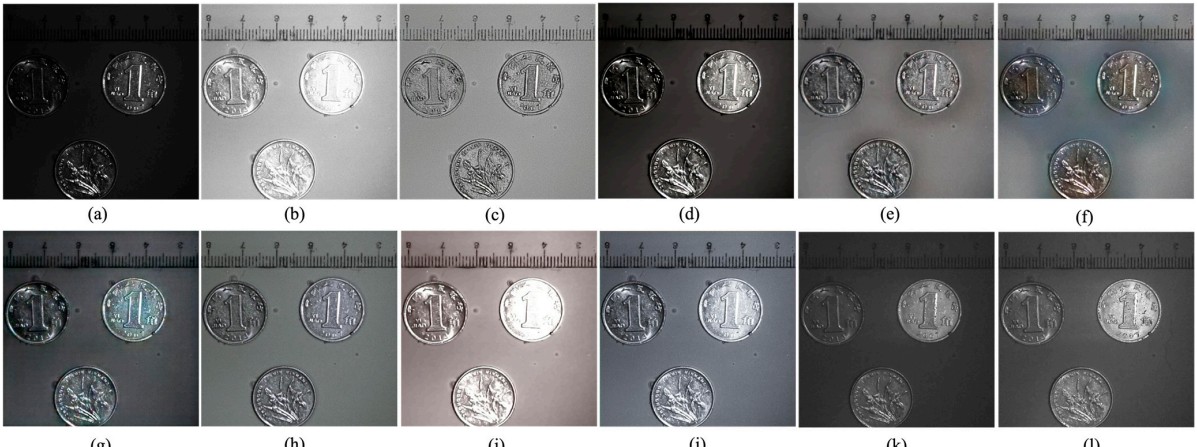

**Figure 17.** Contrast of $S_0$ intermediate enhancement results: (**a**) input; (**b**) ground truth; (**c**) Retinex-Net; (**d**) MBLLEN; (**e**) KinD; (**f**) EnlightenGAN; (**g**) DSLR; (**h**) ZeroDCE; (**i**) RUAS; (**j**) SCI; (**k**) Polar; and (**l**) ours.

It is worth noting that there is no standard definition of "ground truth" for low-light enhancement methods, therefore, we do not calculate quantitative evaluation indicators. The "ground truth" shown in the figure is just used to illustrate that under superior brightness conditions, real-world images have different brightness distributions. From the experimental results, many low-light enhancement methods aim to improve the overall visual effect and achieve brightness balance, such as Retinex-Net and ZeroDCE balanced different brightness distributions, although visually it looks more balanced, but destroys the brightness distribution relationship, which will interfere with the calculation of Stokes parameters. This is because low-light enhancement methods are fundamentally different from our purpose of brightness enhancement. The goal of intensity domain low-light image enhancement methods is to obtain the result of visual pleasure, but the purpose of our brightness enhancement is only the intermediate process, the goal is to restore the three Stokes parameters to calculate high-quality DoLP, so this part of the purpose is to restore better brightness distribution, rather than get the optimal balanced visual effects. KinD and EnlightenGAN introduced the artifacts in the process of brightness enhancement, there are also obvious artifacts in the final results shown in Figures 15 and 16. In detail, the MBLLEN method and DSLR method fail to restore details well, resulting in a structural collapse in the final results. RUAS method, SCI method, Polar method, and our method can adaptively adjust brightness for different exposure regions.

Next, we continue to discuss the intermediate result of the Detail Enhancement Network. First, we need to emphasize that in our proposed paradigm, the purpose of the Detail Enhancement Network is to restore the edge information of parameters $S_1$ and $S_2$ rather than brightness recovery. Figures 18 and 19 show the enhanced results of $S_1$ and $S_2$, respectively. From the results, the purpose of the intensity domain low-light image enhancement method is to obtain better brightness improvement. It can also be seen from the results that they perform well in terms of their purpose, but considering only the restoration of brightness without considering the constraint relationship between different parameters will lead to the destruction of polarization information in the final results. The overall brightness can be restored, but the DoLP obtained by calculation loses important detailed structural information. As for the Stokes-domain low-light image enhancement method, Polar also focuses on restoring the detailed structural information of $S_1$ and $S_2$, but from the results in Figures 18 and 19, we can see that thanks to the design of our gradient residual dense feature extraction module (GRD), we have better restoration of edge information, with a more complete overall contour. In particular, it can be seen from Figure 19 that although the overall noise is severe, both the "scale" and coin's edge information can be well restored. This enables us to achieve the best enhancement performance in the final results.

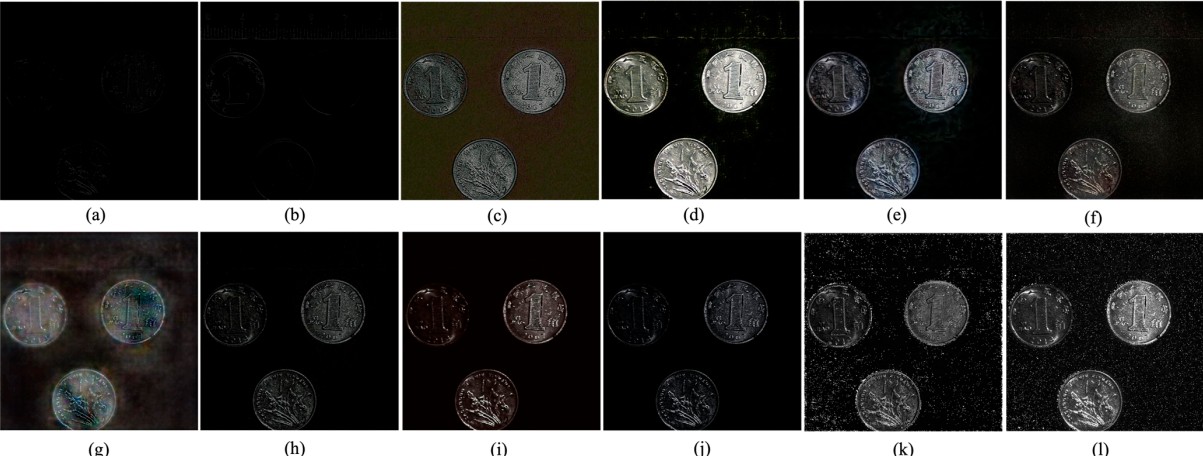

**Figure 18.** Contrast of $S_1$ intermediate enhancement results: (**a**) input; (**b**) ground truth; (**c**) Retinex-Net; (**d**) MBLLEN; (**e**) KinD; (**f**) EnlightenGAN; (**g**) DSLR; (**h**) ZeroDCE; (**i**) RUAS; (**j**) SCI; (**k**) Polar; and (**l**) ours.

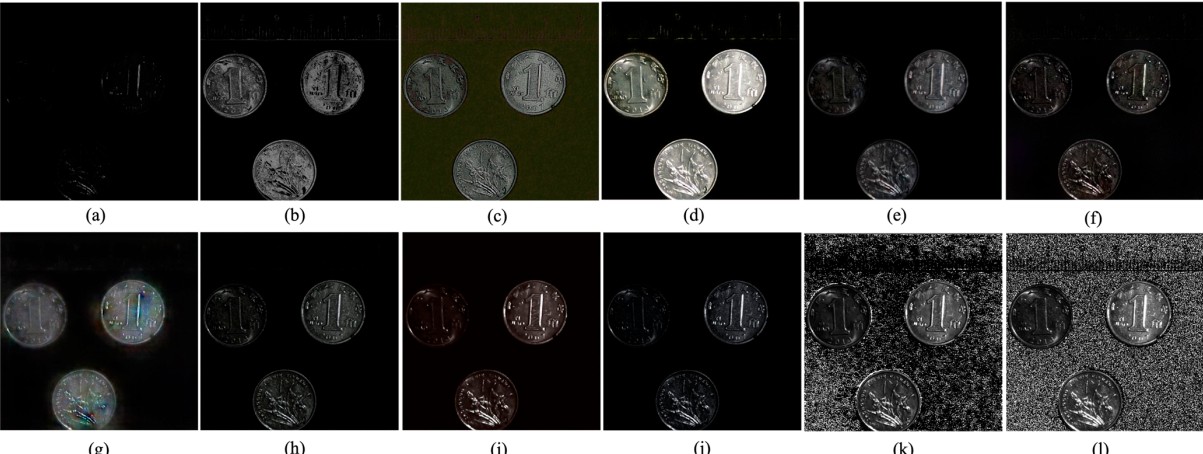

**Figure 19.** Contrast of $S_2$ intermediate enhancement results: (**a**) input; (**b**) ground truth; (**c**) Retinex-Net; (**d**) MBLLEN; (**e**) KinD; (**f**) EnlightenGAN; (**g**) DSLR; (**h**) ZeroDCE; (**i**) RUAS; (**j**) SCI; (**k**) Polar; and (**l**) ours.

From the discussion of the above intermediate process results, we prove that the proposed Stokes-domain underwater low-light polarized image enhancement paradigm is very effective. The dual-branch network realizes different functions, respectively. Compared with the intensity-domain low-light image enhancement method, we define different enhancement tasks that are proven to be very effective. Compared with the Stokes-domain enhancement method, the special network module designed can better realize the brightness adjustment task and the target edge restoration task, proving the superiority of our network design.

### 4.3.3. Quantitative Analysis

We aim to recover high-quality DoLP in the low-light underwater environment, and the purpose of the restoration is to improve the brightness and detail information of the image, mainly from the structural stability and image detail recovery evaluation method performance. Therefore, we select PSNR (Peak Signal to Noise Ratio) and SSIM (Structural Similarity Index) [43] as the evaluation index [44]. SSIM is used to quantify the structural similarity between two images, and PSNR is used to measure the quality of the enhanced image. Since the recovery effect of the denoising method is too poor, the calculated PSNR and SSIM values have no reference value, so we only show the comparison results of the low-light enhancement method. We calculate the average of PSNR and SSIM in the test set, and the results of the real-world experimental are presented in Table 1, from which we can see that our results are superior to all other methods in terms of PSNR and SSIM. Polar obtains inferior results.

**Table 1.** Quantitative comparison on real-world datasets. The best results are bold-underlined.

| Metrics | Retinex-Net | MBLLEN | KinD | EnlightenGAN | DSLR |
| --- | --- | --- | --- | --- | --- |
| PSNR↑ | 9.8347 | 5.7199 | 10.2895 | 14.4324 | 6.7146 |
| SSIM↑ | 0.2041 | 0.1516 | 0.2718 | 0.2410 | 0.2446 |
| **Metrics** | **ZeroDCE** | **RUAS** | **SCI** | **Polar** | **Ours** |
| PSNR↑ | 17.2694 | 15.8941 | 17.0511 | 20.9472 | **24.9282** |
| SSIM↑ | 0.1992 | 0.3799 | 0.2938 | 0.3241 | **0.4674** |

### 4.4. Ablation Study

To demonstrate the effectiveness of the individual components of our method and the effectiveness of our proposed paradigm based on Stokes parameters constraint design, we perform a series of ablation studies, the results of which are shown in Figure 20.

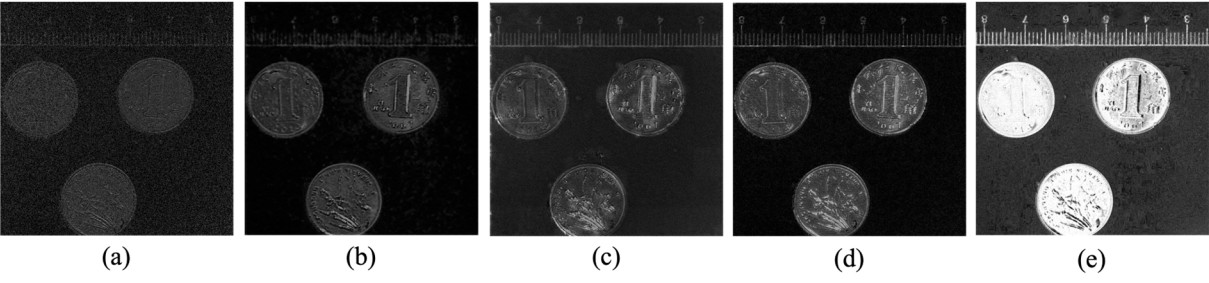

**Figure 20.** Comparison of visual results of ablation experiments: (**a**) input; (**b**) Single-Branch; (**c**) *w/o GRD*; (**d**) *w/o* $L_{polar}$; and (**e**) full method.

First of all, we compare with the model of estimating parameters $S_0$, $S_1$ and $S_2$ with a single branch to verify the effectiveness of our proposed paradigm, which is denoted as Single-Branch, and the experimental results show that the image brightness recovery effect of Single-Branch enhancement is not ideal, and the detail recovery effect is not good, even if we design a very effective feature extraction module, it still cannot maintain the stability of the structure, and it needs to rely on the constraints of polarization relationship to recover

the structural details well. Secondly, we remove the gradient operator and chunk-self-attention module in the gradient residual dense feature extraction module, and only use the dense connection, which is denoted as *w/o GRD*, and the experimental results show that the edge discontinuity occurs, which proves that the complete feature extraction module proposed can extract the edge information in the polarized image well. Finally, we also verify the significance of the loss function by removing the polarization loss function $L_{polar}$, which is denoted as *w/o $L_{polar}$*, and the experimental results of removing the polarization loss constraint are average in detail recovery. The corresponding quantitative evaluation indicators are given in Table 2, and the results of the ablation experiment show that our complete model achieves the best performance.

**Table 2.** The results of the ablation study. The best results are bold-underlined.

| Metrics | Single Branch | *w/o GRD* | *wo $L_{polar}$* | Full Method |
|---|---|---|---|---|
| PSNR↑ | 22.8461 | 21.8611 | 22.0593 | **24.9282** |
| SSIM↑ | 0.3392 | 0.3146 | 0.3166 | **0.4674** |

## 5. Conclusions

In this work, we propose a Stokes-domain underwater low-light polarized image enhancement paradigm based on the physical characteristics of Stokes parameters found in experimental observations and design a dual-branch network based on the paradigm. To drive the network training, we also propose an underwater low-light polarization image simulation method and construct a simulation dataset containing various water types and scene information. To verify the effectiveness and robustness of our method, we set up an underwater polarimetric imaging system and test the performance of our algorithm on the captured real-world dataset, and the experimental results prove that our method achieves the best performance. We also demonstrate the effectiveness of our proposed components and the effectiveness of our design based on the physical properties of the Stokes parameters through a series of ablation experiments. Our work provides a new way to study low-light polarization imaging enhancement. Of course, our work still has some shortcomings, which are currently limited by our research equipment which can only acquire gray images. In the future, this work will also be extended to the field of color polarized images. In addition, we will further explore the possibility of polarization imaging in more scenarios and continue to explore how to combine polarization prior information in the network to achieve better detection results.

**Author Contributions:** Conceptualization and methodology, C.X.; validation, C.X. and Q.L.; writing—review and editing, C.X. and Q.L.; supervision, Y.H. and F.Y.; funding acquisition, E.C. and F.Y. All authors have read and agreed to the published version of the manuscript.

**Funding:** This work was supported in part by the National Natural Science Foundation of China (62271425, 62371404), the Xiamen Ocean and fishery Development Special Fund Project (23CZB004HJ01) and FOCAL project (No. FOCAL2023-0103).

**Data Availability Statement:** The data presented in this study are available on request from the corresponding author. The data are not publicly available due to privacy.

**Conflicts of Interest:** The authors declare no conflict of interest.

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
