# Peer review of "A Dual-Branch Autoencoder Network for Underwater Low-Light Polarized Image Enhancement"

_remotesensing, doi:10.3390/rs16071134_

Round 1
Reviewer 1 Report
Comments and Suggestions for Authors
This manuscript proposes a Stokes-domain underwater low-light polarized image enhancement paradigm based on the physical characteristics of Stokes parameters and designs a dual-branch network based on the paradigm. According to the experimental results, compared with existing methods, there has been a great improvement. In the design of the network, the combination of physical characteristics enhances the interpretability of the network, which shows good innovation. However, there are still some problems that need to be modified in the article. The specific suggestions are as follows:
1. In the manuscript, there are inconsistencies in the expression of "polarization image". Please unify polarization image and polarized image. And distinguish between singular and plural forms uniformly.
2. In the Equation36 for polarization loss, DOP seems to refer to DoLP mentioned in the previous paragraph. However, this acronym was first used without giving its full name, causing confusion among readers. For DOP, please add the explanation in the manuscripts.
3. The Gradient Residual Dense Feature Extraction Module (GRD) mentioned in the manuscript should be clearly indicated in Figure 7, or an additional network structure diagram should be given.
4. Page6 mentions that the linear polarization component is used in the manuscript, so why not use the circular polarization component to fully express the Stokes parameter?
5. There are some errors in the division of the block diagram of Figure 8 in the manuscript, please revise it carefully and replace it with a clearer version.
6. The English expression of the manuscript should be further polished, especially some capitalization errors, such as “Noise2noise” in the legend of Figure 14 in Page16, while “Noise2Noise” is used in the previous paragraph.
7. The intermediate results given in Fig. 18 and Fig. 19 do not seem to perform better than the methods in the intensity domain in terms of human visual effects, but the methods in the polarization domain are much better than the methods in the intensity domain in the final results. Please explain further.
Comments on the Quality of English Language
The English expression of the manuscript should be further polished, especially some capitalization errors, such as “Noise2noise” in the legend of Figure 14 in Page16, while “Noise2Noise” is used in the previous paragraph.
Reviewer 2 Report
Comments and Suggestions for Authors
In this work, the authors propose a Stokes-domain underwater low-light polarized image enhancement paradigm based on the physical characteristics of Stokes parameters found in experimental observations and design a dual-branch network based on the paradigm. To drive the network training, the authors also propose an underwater low-light polarization image simulation method and construct a simulation dataset containing various water types and scene information. Experiments demonstrate the effectiveness of proposed components and the effectiveness based on the physical properties of the Stokes parameters. This paper is well-organized. The experimental results and discussions are relative rich. Here are some points for review and suggestions:
1. The abstract section of the paper is recommended for rewriting. Generally, descriptions of parameter symbols are not beneficial to include in the abstract. Such as S0, S1,…
2. English expression needs improvement, there are some typos and grammar mistakes. It needs an English native speaker for language polishing.
3. The literature format needs to be standardized according to the requirements of the journal. In addition, most references are 3 years ago. Some related literatures in recent years are recommended. For example: ‘Low-light image enhancement based on virtual exposure, 2023’ and ‘Low-light image enhancement by deep learning network for improved illumination map, 2023’.
4. How about the computational complexity of the new method?
5. Is there any limitations of the proposed method?
6.Only few images are used for testing, more samples will be better.
7. In Fig.6, The architecture of Brightness Adjustment Network is not clear enough.
8. The motivation of the method should be discussed.
9. The main contributions are summarized, however, the sentences in the paragraph are too long. It's recommended that each highlight be supplemented with over 30 words.
Comments on the Quality of English LanguageNeeds improvement
Reviewer 3 Report
Comments and Suggestions for Authors
This manuscript proposes a highly novel method for enhancing low-light underwater polarized images, effectively utilizing the antagonistic characteristics of Stokes parameters as a prior for the network. The experimental results demonstrate significant improvement over existing methods, and the incorporation physical properties into the network design enhances its interpretability. However, there are areas requiring revision, with specific suggestions as follows.
1. There are some errors in English expression, such as the definition of polarized images. Please standardize the use of "polarization" and "polarized". In Page9, $ I_{\alpha} $ should be revised as $ I_{\alpha i} $,such as Page6.
2. The experimental results presented in the manuscript are in grayscale, yet color is an important aspect of underwater imaging. Please discuss the potential of extending this work to the RGB space as mentioned in the conclusions.
3. The construction of the dataset is one of the significant contributions in the manuscript, but its description should add more details, such as diversity in lighting conditions and types of water. Additionally, please provide an explanation of how the training and testing datasets are divided.
4. For improved reader comprehension, please clarify what the black arrows in Figure 6 (e.g., the bidirectional arrow between $ R_h $ and $ \bar{R}_h $ ) specifically denote.
Comments on the Quality of English LanguageIn general, English grammar and sentence structure are correct, but there are some spelling mistakes. To the goals and results of the study are clear to the reader, please check and revise carefully before submission.
